# Bio-Inspired Morphing Tail for Flapping-Wings Aerial Robots Using Macro Fiber Composites

Vicente Perez-Sanchez [†], Alejandro E. Gomez-Tamm [†], Emanuela Savastano, Begoña C. Arrue * and Anibal Ollero

GRVC Robotics Laboratory, University of Seville, Avenida de los Descubrimientos S/N, 41092 Seville, Spain; vpsanchez@us.es (V.P.-S.); agtamm@us.es (A.E.G.-T.); emasav2@us.es (E.S.); aollero@us.es (A.O.)
* Correspondence: barrue@us.es
† These authors contributed equally to this work.

**Featured Application: The work intends to demonstrate the feasibility of applying different materials and technologies with the aim of performing morphing to different parts of an aerial system for increasing their capabilities and performance.**

**Abstract:** The aim of this work is to present the development of a bio-inspired approach for a robotic tail using Macro Fiber Composites (MFC) as actuators. The use of this technology will allow achieving closer to the nature approach of the tail, aiming to mimic a bird tail behavior. The tail will change its shape, performing morphing, providing a new type of actuation methodology in flapping control systems. The work is intended as a first step for demonstrating the potential of these technologies for being applied in other parts of the aerials robotics systems. When compared with traditional actuation approaches, one key advantage that is given by the use of MFC is their ability to adapt to different flight conditions via geometric tailoring, imitating what birds do in nature. Theoretical explanations, design, and experimental validation of the developed concept using different methodologies will be presented in this paper.

**Keywords:** aerial robotics; UAS; bio-inspiration; soft robotics; MFC; ornithopters; mocap system

## 1. Introduction

The use of Unmanned Aerial Systems (UAS) has increased exponentially in the last decade, including filming, surveillance, transportation, and inspection. Several approaches have been developed over the years for visual inspection [1–3]. In the last years, UAS have been used for tasks involving physical interactions, adding actuators, and different end-effectors for performing manipulation or contact inspection, as can be seen in papers like [4] or [5]. Entire books have even been published involving this matter [6].

Multirotor systems are used in most of the above applications. However, the flight endurance is small and the rotors can be dangerous for persons or valuable objects nearby.

Fixed-wing systems have been successfully used for surveillance and large infrastructure inspection, including power-line inspection [7,8]. They have proven reliability and large flight endurance, which makes them a better choice for long-range inspections. The guidance accuracy of unmanned fixed-wing systems has been improved [9]. However, fixed-wing systems have difficulties in performing low-speed flights for inspection and lack the maneuverability and vertical take-off and landing properties that are required for many applications.

In recent years, unmanned flapping-wing system, or ornithopters, have started to gain in prominence. These bio-inspired systems offer advantages, like in-flight endurance, when comparing to multi-rotors. Additionally, they are able to glide to save energy, although they have better maneuverability than fixed-wing systems, performing low-speed flight for inspection and being able to perch in very constrained surfaces.

Nevertheless, strict payload limitations do exist in all of the above systems. This has as a consequence in that the miniaturization of some systems, for performing some required tasks in some specific locations, has a limit when using traditional materials and actuation methods.

Therefore, developing a flapping-wing robot that can be used for surveillance, inspection, and emergency transport, as well as being able to perform autonomous flight, with minimal or no intervention of human pilots, requires optimization in weight and energy consumption.

This work proposes using Macro Fiber Composites (MFC) for developing a bio-inspired tail. Normal ornithopters tails need two servo motors for performing the movements of the tail to control roll, pitch, and yaw. The use of MFC will remove these servos and they will enable performing morphing of the tail without them.

The MFC is a type of smart material, which is classified as a piezoceramic. The actuation of this material is based on the piezoelectricity, which was first studied by Pierre and Jacques Curie in 1880. They discovered changes in the shape of different materials when they were under the actuation of an electric field. These changes of the shape, properly produced, could cause a displacement.

This work is based on the application of the piezoelectricity to produce displacements. Moreover, the displacements of piezoceramics materials can also be measured due to the high sensitivity of their electrical properties to deformation. The aim of this work is to perform continuous displacements in the tail of an ornithopter. When combining it with the measurements of these displacements, the flapping-wing flight of the ornithopter will be controlled.

Some previous work-related to the UAS and the MFC will be now presented and discussed.

### 1.1. UAS Actuation State of Art

Miniaturization and weight optimization have been explored in multi-rotors. Approaches, like [10] or [11], show the interest of developing low weight actuators, optimizing payload, using low weight materials, and very efficient mechanical actuators. These approaches have given favorable results in terms of reducing weight and being capable of using a smaller and less dangerous platform. Even so, more miniaturization would reduce the capabilities of these systems. Therefore, a mechanical limit has been reached in these terms.

Conventional servomotors, used in all previous applications, impose limitations in terms of weight and efficiency. Some of the authors involved in this work have researched various alternative materials for performing actuation, trying to obtain the highest force/weight ratio, and removing traditional components from the aerial systems. In [12,13], some potential materials, which could be used as actuators, are presented and detailed. In [14,15], the application of a soft landing system for multirotors using 3D printed soft materials is shown.

There are various materials with high potential for being used in these types of applications. For example, Twisted and Coiled Polymers (TCP), as seen in [13], or shape memory alloys have the potential for being used as a closing mechanism in aerial manipulation.

As pointed out above, flapping-wings appears to be an alternative to multirotors and fixed wings. However, payload limitations are very relevant. Hence, all of th weight reduction that can be achieved is positive.

Additionally, flapping-wings devices opened the field of bio-inspired approaches taking inspiration in animals, like birds or bats. The idea is to optimize the behavior of these aerial systems mimicking animals, as nature tends to optimize their creatures in terms of energy consumption and efficiency.

Flapping-wings properties have been studied over the years in terms of aerodynamics [16] or control algorithms, using even deep learning approaches [17]. They have been applied for inspection tasks, using them to stream video images [18].

Additionally, some approaches removing mechanical parts for piezoelectric actuators have been done. Some examples are [19,20], where flight, and even takeoff, using these actuators are proven.

Nevertheless, the above approaches used the micro aerial system of very low weight and size for their experiments and validation. Making these approaches scalable to medium-sized flapping-wings systems is a great challenge.

Therefore, this work takes a medium-sized flapping-wings system analyzed in [21] as a model. This commercially available system is interesting due to its robustness and modularity.

For designing the tail, biological inspiration is taken based on previous work of bird tail studies. Tails complexity comes to its greatest exponent in sexually dimorphic species, as can be seen in [22]. However, these designs do not improve the performance of these species in flight. They are conceived to improve the sexual selection of males with longer tails. On the other side, a more simple tails' geometrical design can improve several aspects of fly efficiency, like maneuverability. Focusing on these types of tails in [23], it is shown that the design that reduces the aerodynamic cost to the minimum is shallow and deep fork tails. Therefore, the design developed in this work should take advantage of these conclusions, as energy efficiency is one of the key aspects of this work.

The idea is to develop the tail for the integration in an ornithopter especially designed for the GRIFFIN project and still on development [24]. Figure 1 presents a prototype of this design.

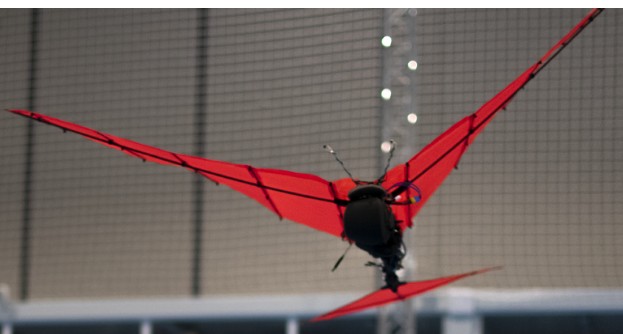

**Figure 1.** Prototype of Ornithopter, Named "Powerbird", Developed for the GRIFFIN Project.

The idea is to achieve a bio-inspired morphing of the developed tail using MFC. By doing so, it will be possible to get rid of the two servos and the mechanical components normally used to obtain this kind of movement. The weight will be reduced, and a more natural and smooth motion will be obtained. The energy consumption will be also reduced, as the MFC consumption is very low. As a result, optimization in various aspects will be obtained.

This is a challenging objective, as displacement in both directions should be achieved with enough movement for being significant while generating enough blocking force. This study could open the possibility of performing morphing in other parts of the bird for optimizing behavior.

The MFC state of the technology is summarized in the following.

### 1.2. Macro Fiber Composites

MFC were developed by NASA in the late 90s for controlling vibration, noise, and deflections in different structures due to their capability of sensing low displacement in their structure, which can be measured as a tension variation. Thus, the first works that can be found are related to structural control applications [25].

Additionally, MFC started to be mentioned in piezoceramic surveys in the early 2000s [26]. In [27]; the properties of these piezoceramics for sensing, actuating and power generation were analyzed. Thus, the research about these MFC started to change from their use as a precise and low-cost sensor to a low weight actuator with high accuracy and

blocking force. Even with their relatively low displacement, these actuators offer a very high force/weight ratio. Therefore, some new potential applications appeared.

Some works started to look for potential applications in Unmanned Aerial Vehicles (UAV) [28]. However, one of the first practical applications as an actuator on robots was for a bio-mimetic fishtail [29], due to the lower frequency of the movement and the smaller amount of force needed to control the robot displacement in water.

Similar publications developing underwater robots were also published at the same time [30–32]. All of them highlighted the potential of the MFC in the fields of low weight, high force, good displacement, precision, and flexibility. Therefore, they were able to mimic the movement behavior of different submarines creatures and reach not only forward motion, but also whirling and multiple direction displacement while using several MFC.

The next step in MFC development was to try them outdoors in unmanned aerial robots. In small devices, wing actuation was obtained by using MFC in fixed-wings [33] or even in flapping-wings [34]. In the same way, [35] used MFC to perform morphing on a wing to change their profile. In the same year, other works were published, like [36], where the potential of the use of MFC in aerial systems was shown. In that work, they designed a fixed-wing aerial robot that was capable of using MFC for actuating the flaps of the wings and tail.

The above publications pointed out the great potential of the MFC for actuating robotic parts that were normally actuated by mechanical systems. The high speed of the displacement and high blocking force were the main reasons. Therefore, some of the works began to use these actuators to drive the fins of robotic fishes [37].

In aerial vehicles, the possibilities of these MFC for active flight control in Micro Aerial Vehicles (MAV) [38] were studied, validating the strength of the actuators and trying to maximize the effective roll and pitch. Composite materials were used, allowing passive load alleviation. Continuing with aerial systems, the suppression of tail buffeting using MFC has also been demonstrated [39], taking advantage of the high strain that these actuators can offer.

The work presented in this paper aims to offer a novel solution, developing a new approach to a morphing tail using MFC. Recent works, like [40], have tried to perform morphing of bio-inspired tails made out of elastomers. They achieved up to 12mm of displacement and validated the concept in a wind tunnel, with the idea of using it on bio-inspired fixed and flapping-wings designs.

The rest of the paper is structured, as follows. Section 2 will focus on the model validation, explaining the MFC properties and theoretical analysis. Section 3 will present the tail concept and design, the attributes of the tail will be explained, the design and development will be described, and the final model will be presented. Section 4 will then present the experimental validation, showing the setup used, the tests performed, and the final results. Finally Section 5 will show the conclusions and future works.

## 2. MFC Tail Actuator Characterization

The approach that is presented in this work aims to develop a bio-inspired tail, where the MFC is the structural part of the system, allowing for higher displacement and lower weight while maximizing the blocking force that MFC can apply. Therefore, this concept intends to develop a tail that maximizes the displacement reaching up to several centimeters. It will be shown, with this significant displacement, the potential for using it in aerial system with real flapping-wings.

### 2.1. MFC Actuator Structure

The MFC actuator is composed of four layers of different materials. The center layer is the piezoelectric material, which is the most important one, because it performs the actuation. This layer is sandwiched by the rest.

Surrounding the central layer is the one that is composed of copper electrodes and epoxy. The epoxy guarantees the position of the copper electrodes. These electrodes per-

form the electric field that produces the movements of the piezoelectric material. The orientation of the copper electrodes is crucial, as they are the ones that produce the deformation.

In the case of the MFC used in this work, the electrodes are positioned transversely, which generates a deformation in the longitudinal direction. The uniform distribution of the electrodes generates a continuous displacement.

The next layer is made of an acrylic material that protects and isolates the previous layers.

Finally, there is a layer made out of Kapton. Kapton is a material with excellent thermal properties. It is usually used to isolate the electronic components when soldering is needed. In this case, the terminals of the electrical power should be welded, but, due to the isolation with Kapton on the rest of the surfaces, the MFC can be used in thermal applications. Figure 2 presents an image of the separated layers.

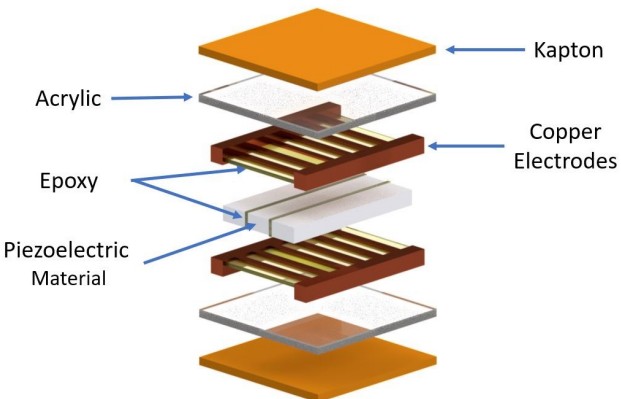

**Figure 2.** Representation of the Separated Layers Normally Present in the Macro Fiber Composites (MFC).

The previously described MFC composition was the one that was developed and patented by NASA. Actually, MFC actuators can not produce vertical displacement by themselves, and they need to be attached to a host material for performing this deformation or morphing. This will be further detailed in Section 3.

The properties and capabilities of the MFC actuator are based on its composition. The different layers add their properties to the final actuator. The MFC selected for this work can actuate in a longitudinal direction. The actuation is based on the electric field that is generated by the copper electrodes and the influence on the piezoelectric material.

In practice, the actuation is based on the properties of passive materials. These materials add stiffness to the actuator, and this stiffness generates internal forces. Accordingly, these forces directly affect the performance of the MFC, generating deformations in different directions, not only in the longitudinal one.

In this work, it is intended to maximize the vertical displacement of the actuator by properly managing the internal forces.

Additionally to the vertical displacement, maximizing the blocking force of the system is also needed. The blocking force opposes the forces of the wind and other external forces, making the deformation stable. A stable actuation is very important in maintaining control over the system. In this work, a balance between the blocking force and the displacement of the actuator is intended. These variables are related with the stiffness of the actuator. This is an equilibrium problem: a major displacement can be achieved by reducing the stiffness of the actuator, but this reduction of stiffness also reduces the blocking force. Several experiments were carried out and the best equilibrium point was chosen, having a good displacement for controlling the robot while maintaining the needed blocking force.

### 2.2. MFC Theoretical Displacement Model

In this subsection, the theoretical model of MFC actuators is presented. This model is based on the classical lamination theory to calculate the properties of the MFC. Subse-

quently, the equations for calculating the theoretical potential energy of the MFC are shown. This is described in more detail in [28].

MFC is composed of several layers that should be individually characterized to calculate the laminate stiffness quantities. The layers can be either isotropic or orthotropic.

In the case of the orthotropic layers, as they have different properties depending on the direction they are measured, they need four independent short-circuit stiffness quantities $E_x$, $E_y$, $v_{xy}$, and $G_{xy}$ for describing their mechanical behavior, where $E_r$ is the elastic modulus in direction "$r$", $v_{xy}$ is the Poisson ratio, and $G_{xy}$ is the shear modulus in the $xy$ plane of the fiber. This is developed in greater detail in [41].

The isotropic layers can be analyzed as orthotropic ones by assuming that $E_x = E_y$ and $G_{xy} = G$.

By using the Classical Lamination Theory (CLT), it is possible to obtain these variables starting from the individual layers and combining them for obtaining the whole MFC.

To achieve this, some assumptions have to be made for transforming the problem from a third-dimensional problem to a two-dimensional one. These assumptions are the following: the displacement is assumed as continuous in the whole laminate; the Kirchhoff hypothesis is assumed; the strain–displacement relationship is linear and the material is linearly elastic, and the through-thickness stresses are small when compared with the in-plane ones.

By doing so, it is possible to obtain the relation of the stress $\sigma$ and strain $\epsilon$ with the following equation:

$$\sigma_k = Q_k \cdot \epsilon_k \tag{1}$$

where $Q$ is the stiffness matrix and $k$ indicates the layer.

Orthotropic layers have greater complexity while calculating their properties as isotropic layers, which produce reduced stiffness matrices. Some of the layers of the MFC are isotropic, like the Kapton or the Acrylic ones, which makes the calculations more simple.

By applying these methods and taking the properties of the layers presented in Section 2.1 into account, it is possible to obtain the stiffness quantities of the whole MFC, as seen in [28]. These values shall be similar to the specifications of the manufacturer.

The calculations to obtain the displacements equations are based on the Rayleigh–Ritz method. This method assumes that the potential energy, depending on the displacement variable, is constant in specific situations. This work focuses on the situation when the actuator is actuated. For a similar development, see [28].

The first step is to specify the displacement equations. It is assumed that the displacement in the mid-line of the actuator and the host lamina are:

$$u^0(x,y) = c_3 x - \frac{c_1^2 x^3}{6} - \frac{c_1 c_2 x y^2}{4}$$
$$v^0(x,y) = c_4 y - \frac{c_2^2 y^3}{6} - \frac{c_1 c_2 x^2 y}{4} \tag{2}$$
$$w^0(x,y) = \frac{1}{2}(c_1 x^2 + c_2 y^2)$$

where $u$, $v$, and $w$ are, respectively, the displacement in the $x$, $y$, and $z$ directions. $c_1, c_2, c_3$, and $c_4$ are the unknown constants.

The next step is to calculate the potential energy assuming the displacement equations. The state when the system is evaluated should be determined in order to calculate this potential energy. In this case, the potential energy is calculated in the actuated state. This is the most complex situation, including the potential energy of the actuated MFC and the potential energy of the host lamina. Therefore, the total potential energy of the system can be expressed as:

$$U_{TOTAL} = U_{HOST}(c_1, c_2, c_3, c_4) + U_{MFC,Actuated}(c_1, c_2, c_3, c_4) \tag{3}$$

where the potential energy of the actuator and the host lamina can be separately calculated by the following expressions:

$$U_{HOST} = \int_{-\frac{L_{x,host}}{2}}^{\frac{L_{x,host}}{2}} \int_{-\frac{L_{y,host}}{2}}^{\frac{L_{y,host}}{2}} \int_{-\frac{H_{z,host}}{2}}^{\frac{H_{z,host}}{2}} [\sigma_{host,x}\epsilon_{host,x} + \sigma_{host,y}\epsilon_{host,y}$$
$$+ \sigma_{host,xy}\gamma_{host,xy}]dzdydx \qquad (4)$$

$$U_{MFC,Actuated} = \int_{-\frac{L_{x,MFC}}{2}}^{\frac{L_{x,MFC}}{2}} \int_{-\frac{L_{y,MFC}}{2}}^{\frac{L_{y,MFC}}{2}} \int_{-\frac{H_{z,MFC}}{2}}^{\frac{H_{z,MFC}}{2}} [(\sigma_{MFC,x} - \sigma_{PZ,x})\epsilon_{MFC,x}$$
$$+ (\sigma_{MFC,y} - \sigma_{PZ,y})\epsilon_{MFC,y} + (\sigma_{MFC,xy} - \sigma_{PZ,xy})\gamma_{MFC,xy})]dzdydx \qquad (5)$$

where $\sigma_{i,j}$ are the stresses of the $i$ element in $j$ direction, $\epsilon_{i,j}$ and $\gamma_i, j$ are the strains in the same nomenclature that $\sigma_{i,j}$, and $L_{k,m}$ are the longitude of the material in k direction. The notation of the elements is *HOST* for host material, *MFC* for the MFC complete actuator, and *PZ* for the piezoelectric material of the MFC.

The process to obtain the strains and stresses in the function of the unknown coefficients of the different elements are well detailed in [28].

Finally, the $U_{TOTAL}$ can be expressed as function of $c_1$, $c_2$, $c_3$, and $c_4$. The next step is to apply the Raygleigh–Ritz method as:

$$\frac{\partial U_{TOTAL}}{\partial c_i} = 0 \qquad i = 1, 2, 3, 4 \qquad (6)$$

Regarding this equation, the values of $c_1$, $c_2$, $c_3$, and $c_4$ are known and the displacement equation can be obtained by replacing the values in (2).

Figure 3 shows the theoretical model deformation of a MFC actuated by 1500 V. The colors represent the deformation in the Z-axis, where the red color is the major deformation. This deformation is consistent with the real deformations in the experiments that will be detailed in further sections.

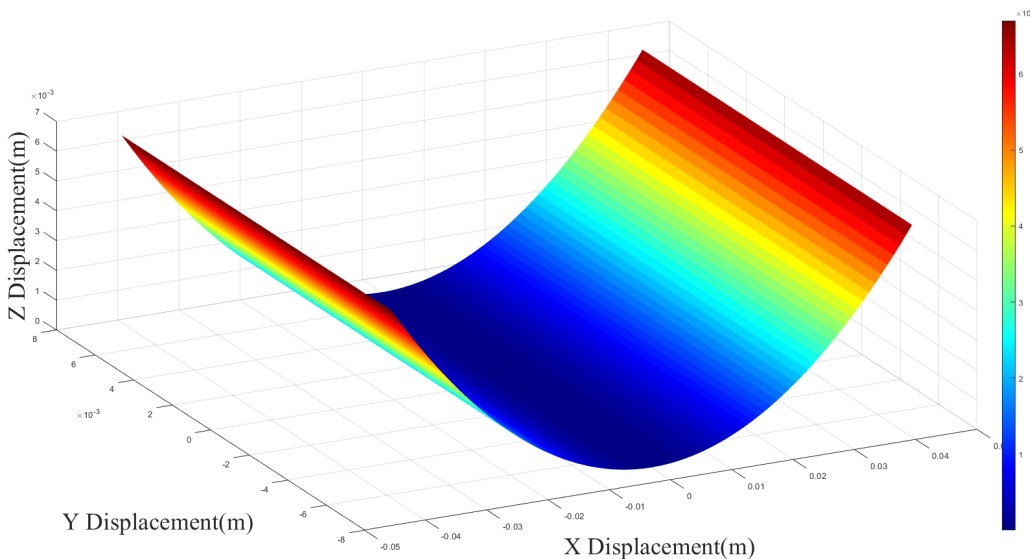

**Figure 3.** Theoretical Model Representation.

Tail Actuators Comparison

MFC present some features that can improve the behavior of these as actuators when compared with traditional solutions. This subsection will focus on explaining the advantages that are offered by MFC when compared with the traditional way of actuating a flapping-wing tail, the servomotor.

The comparison will be done between a commercial SAVOX SH-0255MG servomotor, which is used in the same project for the traditional approach of the tail, and the previously described P1-8514 MFC.

The geometry of both actuators is the first main difference between both systems. The servomotor has a cubic shape of 22.8 × 12 × 29.4 mm. This is a bulky shape in all directions. In comparison, the MFC has a shape of 101 × 20 × 0.5 mm. As can be seen, MFC has a very low height. This characteristic allows the integration of these actuators in much more restrictive geometries, allowing it to merge with the desired surface.

Additionally, it is remarkable to mention the energy efficiency. The one of servomotors is high, around 85 percent. However, the one of the MFC is virtually 100 percent, as the energy consumption of these actuators is almost zero.

The weight is also reduced when using the MFC. A main reason for this is that MFC are used as a structural part of the design. They add stiffness to the system and allow for reducing in other materials, so the design is more optimized in terms of weight when compared with the external actuation performed by the servomotor.

The speed is also an important factor, affecting the capability of controlling the tail. The mechanical reaction of the servomotor is acceptable, but not instantaneous, while the MFC have an almost instant reaction when tension is applied to them, as a result of the properties of piezoelectric to deform when a electrical field is applied.

In terms of isolation, MFC are water and dust resistant, so that they can operate in more hostile environments as the servomotors. This adds versatility to the application. Table 1 summarizes all of the compared parameters.

**Table 1.** Comparison between Servomotors and Macro Fiber Composites (MFC) Actuator.

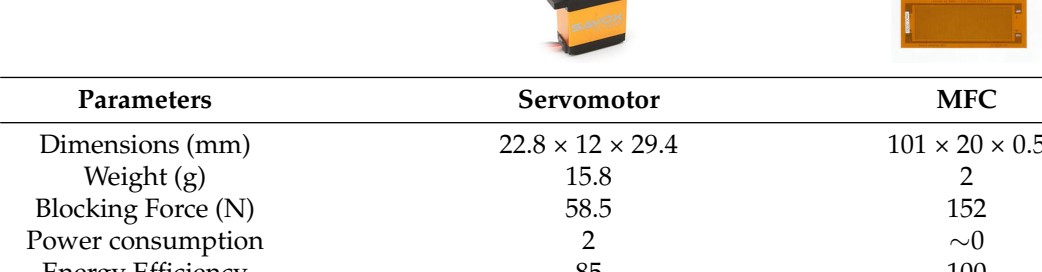

| Parameters | Servomotor | MFC |
|---|---|---|
| Dimensions (mm) | 22.8 × 12 × 29.4 | 101 × 20 × 0.5 |
| Weight (g) | 15.8 | 2 |
| Blocking Force (N) | 58.5 | 152 |
| Power consumption | 2 | ∼0 |
| Energy Efficiency | 85 | 100 |
| Speed | Medium | High |
| Dust and Water resistant | No | Yes |

All of these advantages justify the use of MFC as alternative actuators for performing morphing of the developed bio-inspired tail.

## 3. Tail Concept and Design

In Figure 4, the final prototype of the tail integrated into the platform is presented. The main objective of this section is to explain the reasons and techniques that make the development of the system that is presented in this work possible. This section will focus on the concept and design that enables the development of the desired tail.

### 3.1. Design and Development

The first decision about developing a tail is its geometry. The tail design is similar to the geometry found in nature, as the intention is to imitate animal behavior. Our design has a triangular shape, similar to the one of the commercial ornithopter of the Carbonsail company presented in [21], but with its structural part consisting of two fiber sheets coming from the base. Fiberglass is the material used for the basic structure, due to their low weight and high resistance. However, adding fiberglass in the full length of the tail would

add too much weight and resistance. This would affect the displacement desired to be applied by the MFC.

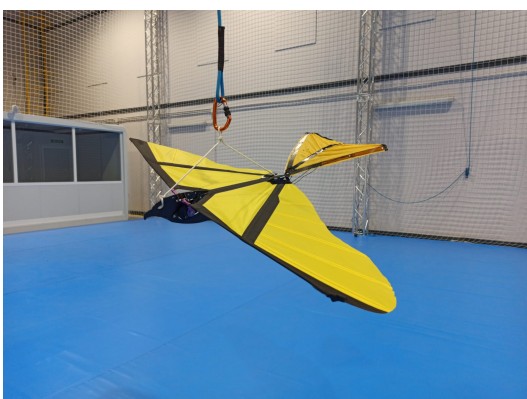

**Figure 4.** Final Prototype Integrated.

Other important restrictions are introduced by the properties of the actuator presented in Table 2. The isolation makes the MFC resistant to water and dust, but the necessity of being attached to a surface implies an increment of weight that should be minimized for a low weight application as the one studied in this paper, as flapping-wings have strict payload limitations. Related to that, as the MFC should be glued to the surface, preferably with epoxy glue, any error in the design of the tail that implies detaching the MFC will lead to weakening the MFC and the appearance of breaks. That will make them useless, as it will create short circuits in the internal piezoelectric structure.

FInally, MFC only allow movement in one specific direction, contracting or expanding, depending on the piezoelectric configuration. In this work, MFC is used for bending a surface when they elongate. Therefore, one MFC can bend one surface only in one specific direction. If one surface shall be bent in both directions, then more MFC should be used in both sides of the surface to allow it.

The MFC used in this design was the P1-M8514, offering a maximal blocking force of 152 Newtons with a total size of 101 × 20 mm and a weight of approximately 2 g. The decision for using these instead of others was based on obtaining the highest displacement with the lowest weight, also offering a good blocking force. Because the priority of this application is low weight and significant displacement, it was considered the best choice.

Therefore, it was decided to place the fiberglass sheet only up to the end of the first MFC. To maximize the deformation, it was discussed setting two or three MFC in series. However, morphing against gravity is the most restricted movement. It is also the most important one, because it allows the system to gain height, while the opposite direction moves the ornithopter in the ground direction.

At last, the decision was to set three MFC in series in the direction against the gravity force and only two in the opposite direction, as the movement in this direction is not as critical. By doing this, the displacement in the critical direction was maximized while maintaining enough displacement in the other direction.

A resistant nylon fiber fabric was the material used as a surface for the tail. MFC were directly attached to the fabric by using epoxy. Therefore, the epoxy, the MFC, the fiberglass, and the nylon fiber fabric formed the complete structure of the tail. It could be directly actuated, performing the morphing and the form of the tail could be kept.

As each row of MFC should be actuated simultaneously, the MFC of the same row were welded in parallel with silicon cables. Therefore, it is possible to use these MFC as if they were one long MFC.

One important issue of trying to achieve a higher displacement placing these MFC in series is that there are no active places between the active layers of the MFC. These spots maintain the shape and they will deform when the piezoelectric performs morphing.

To solve this, the decision was to add rigidity to the bonds, using small fiberglass pieces to guarantee the stiffness of these parts. Because these rigid parts are very short, the morphing maintains their continuity over the whole tail.

Finally, to maintain the stiffness of the tail's fiber, it was decided to fix both sheets with a screw, so that the angle of the tail could be modified. This solution guarantees the rigidity of the tail in the case of manufacturing errors. This is important because the stiffness of the fiber will directly impact the aerodynamics of the tail.

All of the decisions exposed in this section have the idea of developing a bio-inspired system with the most close to nature approach possible, imitating not only the parts of an animal tail, but also its behavior and displacement.

### 3.2. Final Model

After iterating several times, it was decided that the best shape should try to maximize the surface while maintaining a triangular shape that is similar to the one found in nature. The final model is a triangle in which the actuation structure consists of ten MFC for performing morphing. For adding some more rigidity to the system, scotch tapes were added to maintain the tail straight. Figure 5 shows the model.

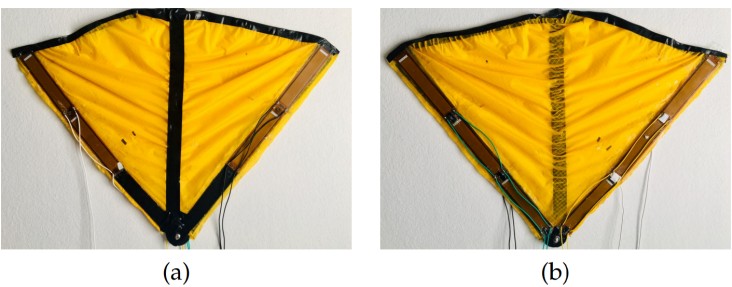

(a)        (b)

**Figure 5.** Final Developed Tail Model Prototype. (**a**) Tail upper site. (**b**) Tail bottom part.

According to [42], different tail configurations can be found in nature to control the bird flight. In Figure 6, the tail developed in this work is presented in the different configurations found in nature for a better understanding. Figure 6a represents the asymmetric YAW control configuration. In this configuration, one actuator of the tail is raised when the other stays at the rest position. This situation generates a moment around the Z-axis of the system to control the YAW. Figure 6b represents the symmetric PITCH control configuration, where the configuration symmetrically activates both actuators generating a moment around the Y-axis of the system to control the pitch. Finally, Figure 6c represents the asymmetric Air-Brake control configuration. In this configuration, both of the actuators are activated proportionally in opposites directions.

For proving the improvement of the developed tail, it was compared with the tail of a commercial Ornithopter. By doing so, the advantages that were obtained when using the morphing tail instead of a classic mechanical actuated tail were shown. Figure 7 shows a classical mechanical tail and the tail developed in this work.

When comparing both tails, one key element is their surface, as it is very important to generate aerodynamics forces. More surface will allow for a better control and aerodynamic behavior when actuating the tail and performing a change of direction. The commercial tail has a surface of approximately 462 cm$^2$, while the developed MFC tail has a surface of 1190 cm$^2$. The weight of the commercial tail is 67.1 g and the one of the MFC tail of 51.8 g, as it can be seen in Figure 7. Therefore, the weight/surface ratio is 0.445 in the MFC tail and 1.45 in the commercial one. This means that this ratio is 3.3 times better in the MFC tail. Because surface is a mayor property for the tail, the conclusion is an improvement of the qualities of the developed tail compared with the traditional approach.

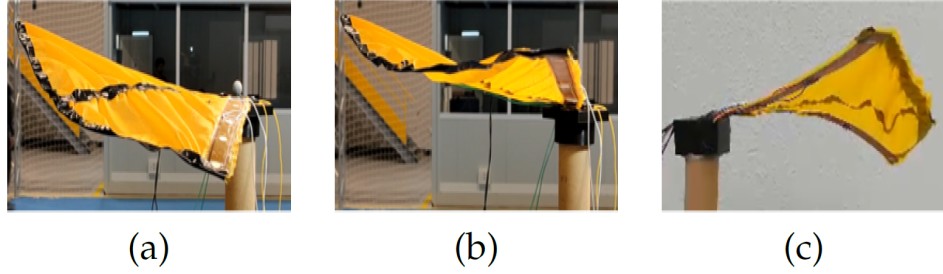

**Figure 6.** Bio-inspired Tail Control Configurations. (**a**) Asymmetric YAW. (**b**) Symmetric PITCH. (**c**) Air-Brake.

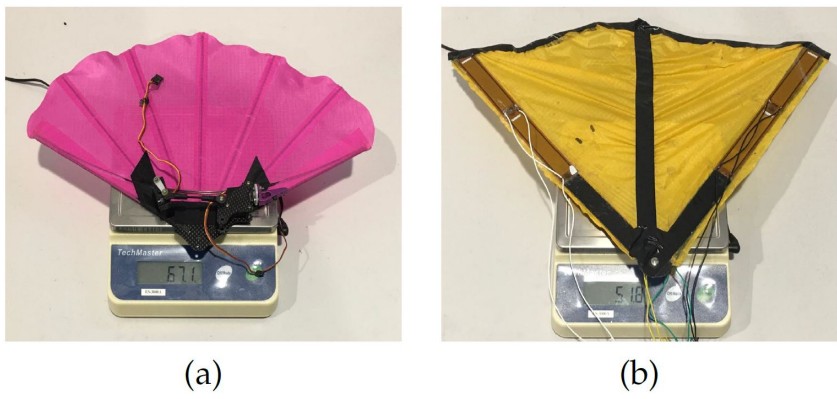

**Figure 7.** Commercial Tail with Servos (**a**) Compared with MFC Tail (**b**).

The displacement performed by the morphing of the tail is also more natural and smoother when compared with the mechanical displacement. Moreover, it allows a compliant movement, as the actuators can change their deformed shape without breaking when an external force is applied. An external deformation of the tail while blocked by the servomotor would cause the servo to break. Table 2 shows a resume of the tail properties comparison.

**Table 2.** Weight/Surface Tails Comparison

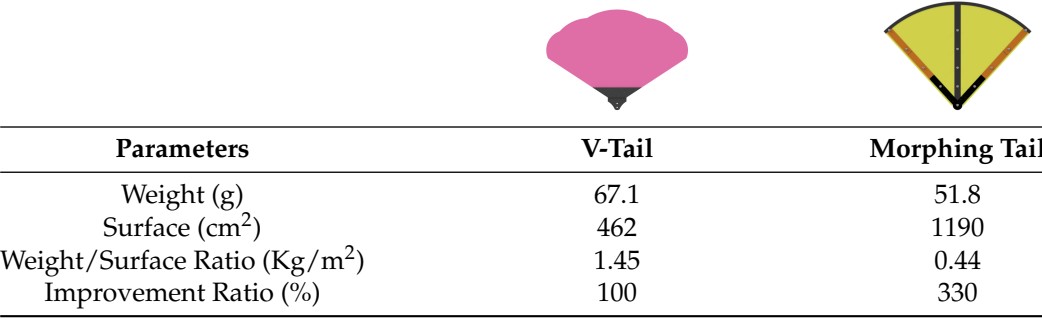

| Parameters | V-Tail | Morphing Tail |
|---|---|---|
| Weight (g) | 67.1 | 51.8 |
| Surface (cm$^2$) | 462 | 1190 |
| Weight/Surface Ratio (Kg/m$^2$) | 1.45 | 0.44 |
| Improvement Ratio (%) | 100 | 330 |

In order to analyze the feasibility of the model, two different experiment configurations were performed several times for proving feasibility and repeatability.

The first one is presented in Figure 8a, where the intention is to raise the tail against the gravity. The second experiment that is presented in Figure 9a tries to analyze the feasibility of the model to perform a twist move in static conditions.

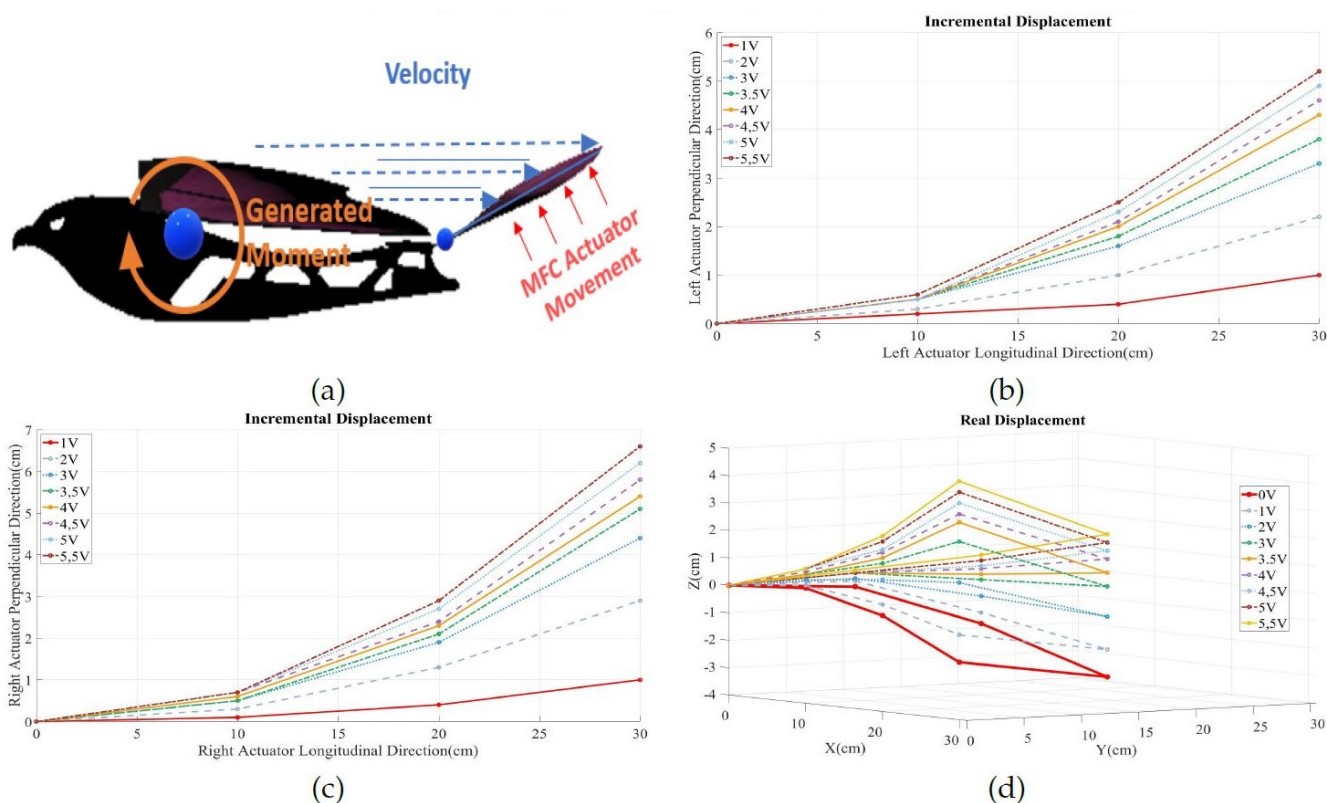

**Figure 8.** Tail Raise Feasibility Experiments. (**a**) Shows the potential behavior of the tail. (**b**,**c**) show the displacement of both the left and right actuators. (**d**) Shows a three-dimensional (3D) representation.

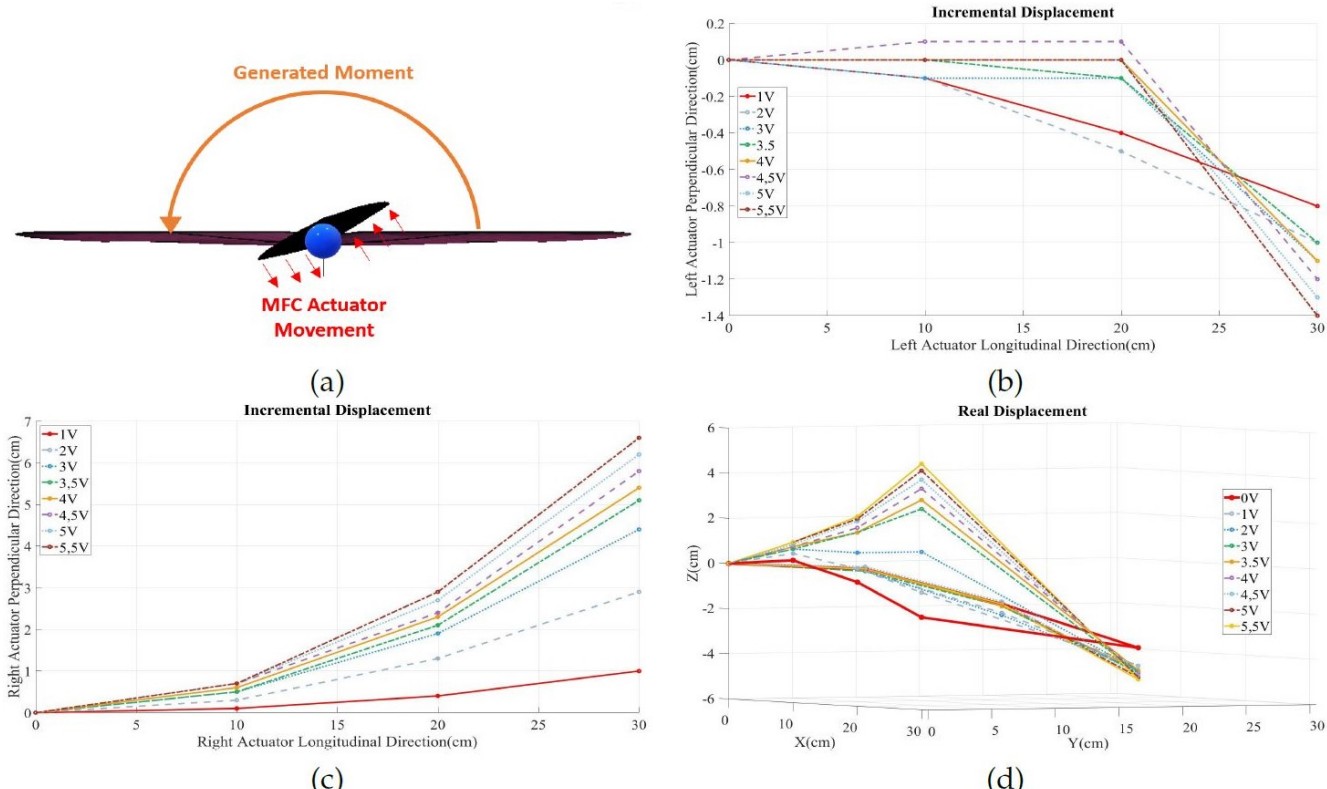

**Figure 9.** Tail Twist Feasibility Experiments. (**a**) Shows the potential behavior of the tail. (**b**) Shows the incremental displacement of the left actuator. (**c**) Shows the incremental displacement of the right actuators. (**d**) Shows the incremental tail twist 3D representation.

Both of the experiments have the same structure on the presentation. Firstly, a figure with a simulation of the experiment is presented. Secondly, the mean of the incremental displacement results on both of the actuators are shown. Finally, the mean of the results of the tail's real displacement referenced to the attachment point are exposed. The experiments have been done in a test-bed in the opposite direction to the gravity forces. The voltage shown in all of the experiments is the input of the DC-DC amplifier.The DC-DC amplifier generates a proportional output from the input values, where 0 V in the input generates 0 V in the output, and 6 V in the input generates 1500 V in the output. Each of the experiments were performed and measured twenty times to ensure reliability.

Starting with the raise experiment, Figure 8a represents the influence of the actuation on the ornithopter trajectory. This deformation generates a moment around the pitch axis that performs a nose up movement.

In Figure 8b,c, the incremental displacement of the tail raise is measured. The incremental displacement of both actuators is similar, as can be seen in the figures. The left actuator loses some actuation, which is due to the handcrafted manufacturing of this model. However, the actuation provided by the model is good. The actuation approximately generates an incremental displacement of nearly seven centimeters in the tip of the tail. This is measured manually while using a measuring tape, so some errors can exist. More precise and dynamic measuring will be performed in Section 4.

Figure 8d represents the real displacement of the tail.The origin of the axis in the figure is the attachment point of the tail to the frame of the ornithopter, and all of the measures are referenced to this point as coordinates origin. The displacement is significant for obtaining a good level of actuation.

The above entails an improvement regarding previous works, because it accomplishes larger displacements. As an example, see [43], in which the displacement achieved is around 1.2 cm. That means the displacement obtained in this work is nearly six times bigger.

Concerning the second experiment, the tail twist is represented in Figure 9a. The tail twist generates a moment in the center of gravity around the roll axis, as can be seen in this Figure. The figure also shows the different actuation on the two parts of the model. The right actuator is rising while the left actuator is falling. This situation generates a twist in the tail which makes it possible to spin around the roll axis of the ornithopter. The Figure 9b–d show the measurements of the deformation in the tail twisting.

Figure 9b,c show the incremental displacement of the actuator. The first graphic shows the incremental deformation on the left actuator and the second graphic shows the incremental deformation on the right one. The deformation on the left actuator is influenced by the deformation on the right one and the tension generated in the fiber by that displacement. However, the position at the end of the actuator also increases with the voltage. Regarding the right actuator, the tension of the fabric does not have an important influence on the movement of this actuator.

Finally, Figure 9d shows a representation on three-dimensional (3D) of the tail movement. Regarding the figure, the actuators can generate an angle of 20 degrees concerning the horizontal plane. The actuators generate displacement enough to generate a moment around the roll axis. Another conclusion of the feasibility analysis is the reliability of the results, which is good. The system has an excellent accuracy in its actuation. The accuracy and repeatability guarantee the actuation and control.

The manufactured model works well. The next section analyzes the capabilities of the tail morphing using a Motion capture (Mocap) system. By using these systems, it is possible to obtain high accuracy measures in more points of the tail and also a major variety of experiments that would be difficult to acquire manually.

### 3.3. MFC Electronics Integration on the Ornithopter

MFC are, as previously mentioned, low-cost piezoelectrics. They allow for changing the shape of the surface they are attached to when an electrical field is applied to them. Accordingly, these properties allow performing morphing with very low energy consumption.

However, some drawbacks should be taken into consideration when using this kind of actuator. Firstly, because high electrical fields have to be applied (up to 1500 Volts), this carries a risk of electrocution if the terminals are not isolated.

Additionally, the behavior of the MFC should be considered. They act as capacitors that get charged by the electrical field applied. Therefore, if the field is removed, the MFC will be still charged and should be discharged by a grounding resistance to allow enough speed in the actuation. Hence, the electronics that are needed to use the MFC must include this grounding resistance.

Thus, it was decided to use a small DC-DC amplifier that could be fed by a 0 to 6 Volts input that could be easily provided on-board of an aerial system. Additionally, low weight and size were very important in the decision. The amplifier and the resistance mentioned can be seen in Figure 10. Moreover, a switch was added for changing the state of the MFC from receiving the tension of the amplifier to be in series with the ground resistance.

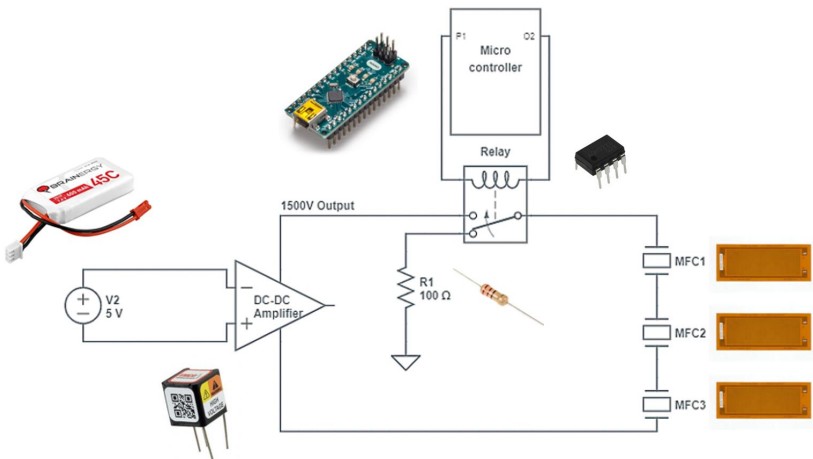

**Figure 10.** Schematic of the MFC Control Board.

Finally, Figure 11 presents the real control electronics system. The weight of the electronics is around 25 gr. It is made on a protoboard to prove the normal operation and the functionality of the system. In the future, the system will be integrated with another control electronics in the ornithopter.

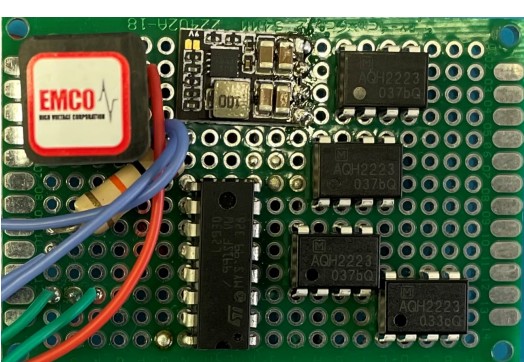

**Figure 11.** Real MFC Control Board.

## 4. Experimental Validation

### 4.1. Experimental Setup

The following experiments were performed within the *Optitrack* System workspace volume, hence the ornithopter tail has been coated, creating a mesh-like distribution of retro-reflective markers.

In an optical motion capture system, the target volume is surrounded by multiple synchronized cameras, 27 *Prime$^x$ 13* [44] in the specific case, which collects 2D images at rates

up to 240 frames per second generating infrared light to enhance markers visibility. After an onboard image processing, cameras identify each marker center with sub-millimeter precision, and then the data is delivered to *Optitrack* software *Motive-Body* [45] that compares all the matching positions and compute three-dimensional data via triangulation process. Finally, the entire markers set motion is represented in a six-degree freedom model, which is obtained inferring rotational information from the relative marker's orientation, as described at [46].

Because of the discussed features, motion capture results in an optimal method to highlight the crucial sections kinematic over the tail span. The use of reflective two-dimensional dots as markers ensures that the entire mesh weight does not affect the results of the experiments. Therefore, this method is particularly indicated when aerodynamic surfaces are involved: e.g., [21] shows the kinematic results that are related to a flapping-wings analysis.

Figure 12a illustrates the whole markers disposition chosen to carry out the tests. As described in Figure 12b, the arrangement of the final markers consists of three radial and five cross-sections including six and three markers, respectively. Cross-sections are created by the lines passing through markers that are located at the same height value and result specifically composed of two symmetrical segments. Figure 12c shows the actuated tail on a test-bench in the Optitrack and Figure 12d shows the *Motive-Body* discretized model during the tail raise cycle. As is shown, the markers density changes along the tail: critical areas, e.g., side corners, need more samples to make their behavior be accurately described.

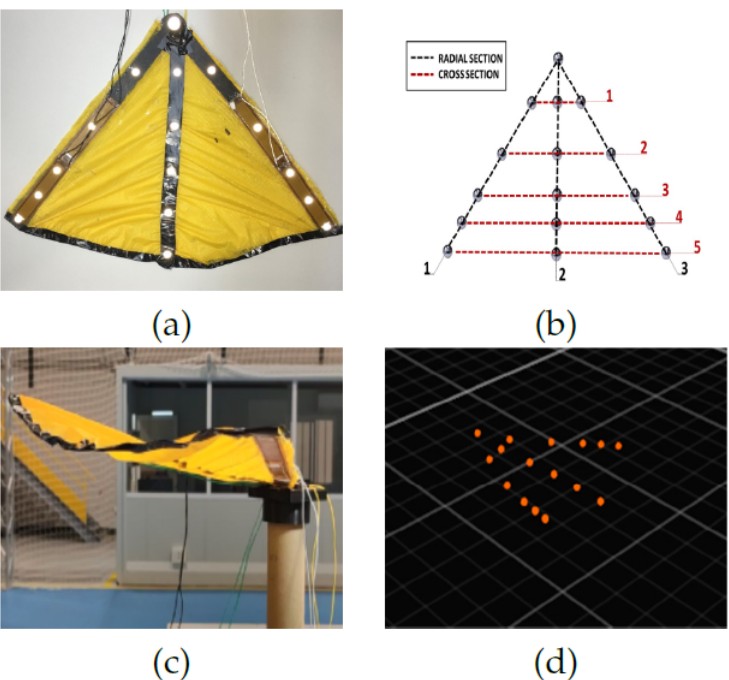

**Figure 12.** Tail Optitrack Representation. (**a**) Shows the markers distribution on the real tail prototype. (**b**) Shows the pattern wanted for performing the mesh. (**c**) Shows the real tail raise view applying 1500 V. (**d**) Shows the markers as seen in the Optitrack.

### 4.2. Kinematic Analysis

As the motion capture system provides the position of each marker over time, an approximated tail deformation analysis is performed.

To highlight the effect of different actuator behavior on the tail kinematic response, the analysis has been carried out when considering an increasing deformation velocity. It is achieved by changing the actuator supply signal over a 1.0–6.0 V range that is turned up to 300–1500 V by the DC-DC amplifier. The results that are related to the tail deformation

analysis are achieved by applying a geometrical approach on both radial and cross directions, depending on the type of solicitations performed. Namely, the following method is conceived to highlight the tail kinematic response observed while both *bending moments* and *torque* rates are generated by actuators.

At first, unstrained references are extracted along each *sections* using the initial position data and creating a polygonal chain across subsequent markers. This method results in the most appropriate to take into account the initial profile curvature even in the cases of e.g., incident air flows at the beginning of the tests. Furthermore, the unstrained reference is necessary to keep track of the deformation of the section concerning the unstrained configurations.

Subsequently, a non-stationary analysis of critical parameters is carried out over the whole sections. For each marker of the polygonal chains, bending and torsion variables are considered to determine the deformation trend.

The *deflection* value $f(t)$ is calculated along the radial directions, while the *torsion angle* $\theta(t)$ is measured along the cross ones, respectively.

Because the tail support has been installed in parallel to the Optitrack reference axis $z$, the value of $f(t)$ for each *n-th* marker has been easily calculated, as described in (7):

$$f_n(t) = z_n(t) - z_{n,0} \tag{7}$$

where $z_{n,0}$ represents the initial $z$ coordinate value that is related to the n-th marker of the considered radial section.

Afterwards, defining $\vec{L_j}(t)$ as the vector associated to the generic cross section segment and pointing to the corresponding side-standing marker, $\theta(t)$ can be evaluated as explained in (8):

$$\theta_{tj}(t) = \arccos\left(\frac{\vec{L_j}(t) \cdot \vec{L_{0,j}}}{\| \vec{L_j}(t) \| \, \| \vec{L_{0,j}} \|}\right) \tag{8}$$

where $\vec{L_{0,j}}$ represents $\vec{L_j}(t)$ initial instance.

The data frequency of the following results is about 120 Hz, so a significant accuracy has been maintained. Figure 13c,f report three-dimensional (3D) *Optitrack* raw data during a single 6 V raise and right-twist cycle.

Figure 13a,b highlight the difference tail response resulting from a bending moment generated with 1 and 6 V supply signals values, respectively. Particularly, each graph reports the behavior of the marker along each radial direction during a single tail raise. As expected, during the 6 V tests, the $f(t)$ value results in almost 8 cm for the tip marker, about 12 times higher than Figure 13a corresponding data. Figure 13d,e instead, illustrate the torsion angle $\theta(t)$ responses that result from the torque applied to the half-right sections of the tail. Those graphs reflect a common torque response for the selected cross directions, that reaches 20 degrees value during a 6 V actuator signal, approximately seven times bigger than the corresponding 1 V data. Figure 13e also stress that $\theta(t)$ value is not affected by the sections height when a strong torque value is applied. The first cross-section is not affected by the torque impulse, so the related results have been omitted, as can be seen in Figure 13f. These results are similar to the one obtained in the previous section, being a little better and more accurate, which confirms the good behavior of the presented tail.

In conclusion, the kinematic analysis highlights the visible increment of tail response velocity. It results in being approximately twice as big during both 6 V raise and torque tests than the corresponding 1 V ones. Furthermore, data demonstrate the tail stability improvement when a higher voltage is applied. As expected, the dynamic analysis presented in this section also confirms the static results that are shown in Section 3.

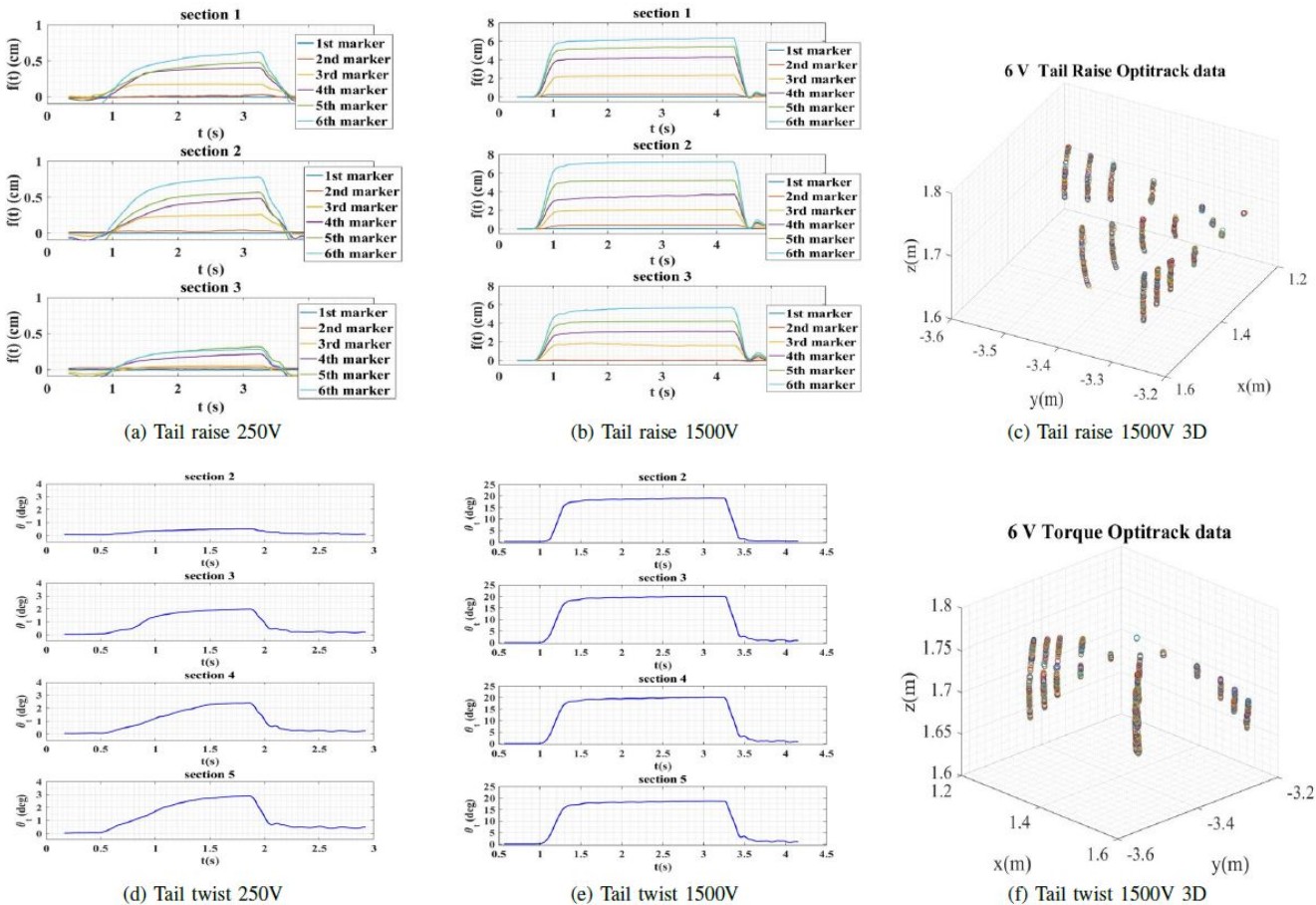

**Figure 13.** Optitrack experiments representation. (**a**,**b**) show the continuous displacement over time with 1 V in the input of the DC-DC converter corresponding 250 V in the actuator input and 6 V in the input of the DC-DC converter providing 1500 V in the actuator. (**c**) Shows a 3D representation using the markers placed on the tail in the Mocap system coordinates. (**d**–**f**) show the same results for the twist experiments.

### 4.3. Repeatability and Control Study

In order to analyze the repeatability of the experiments in wind conditions, the graphics below show the results of an experiment in unsteady wind conditions. The experiment has been performed numerous time for showing repeatability. The lines of different colors represent the voltage of the experiment. The measures of the markers are represented by an interval in which the maximum point represents the maximum of the experiments, the minimum represents the minimum of the experiments, and the mean is represented by a triangle. Figure 14 exposes the tail raise experiments. Concretely the figure shows the measures of the *Radial Section 1* shown in Figure 12b in Z and X Optitrack coordinates. The origin of the graphic is centered in the initial marker of this Section.

The analysis of this graphic provides some conclusions. Firstly, the system is reliable. The accuracy of the experiments measurements is around $+-0.001$ m. This makes it feasible to control the ornithopter trajectory while using the developed system. At the end of the actuator, the uncertainty in the measures is bigger due to the micro-vibrations generated in the material when actuated. The shape generated on the tail is very similar to the one of the bird's tails found in nature. Finally, at 1500 V power supply, the tail achieves 56 degrees of positive deformation; this situation guarantees to have enough actuation to control the ornithopter. As it can be seen, the wind applied induces an increased vibration specially in the tip of the tail. However, the tail is capable of maintaining a considerable deformation even in wind conditions.

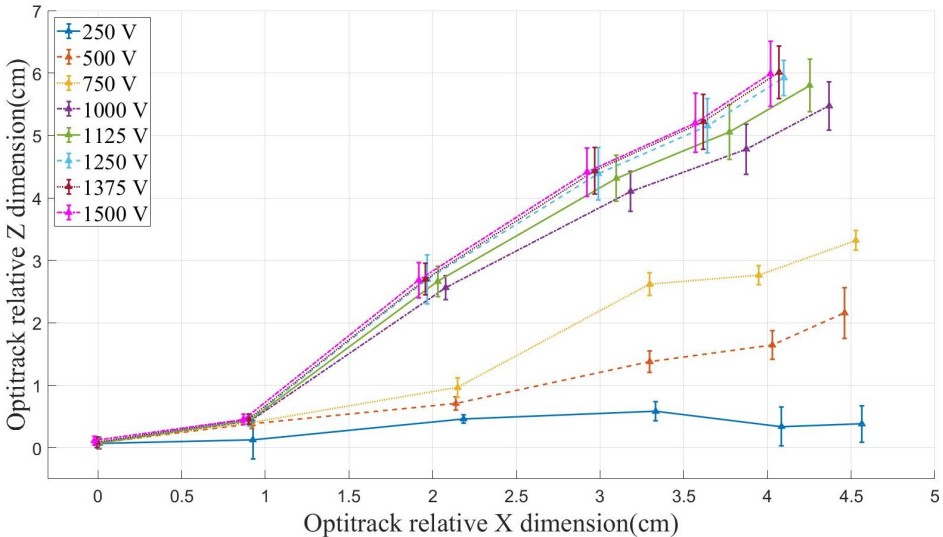

**Figure 14.** Tail raise repeatability experiments: Lateral view.

Figure 15 shows a graphic of the tail torsion experiment performed numerous times for demonstrating repeatability. The range of samples is shown in the graphic with a line in each marker. The representation is similar to the previous one, with the exception of the plain represented and the markers. It is interesting to measure the position of the markers at the end of the actuator in order to analyze the torsion. This graphic shows the position of the *Cross Section 5* represented in Figure 12b. The origin of this graphic is centered in the marker located in the junction of the *Radial Section 2* and *Cross Section 5*. The properties of the representation are the same as the first graphic.

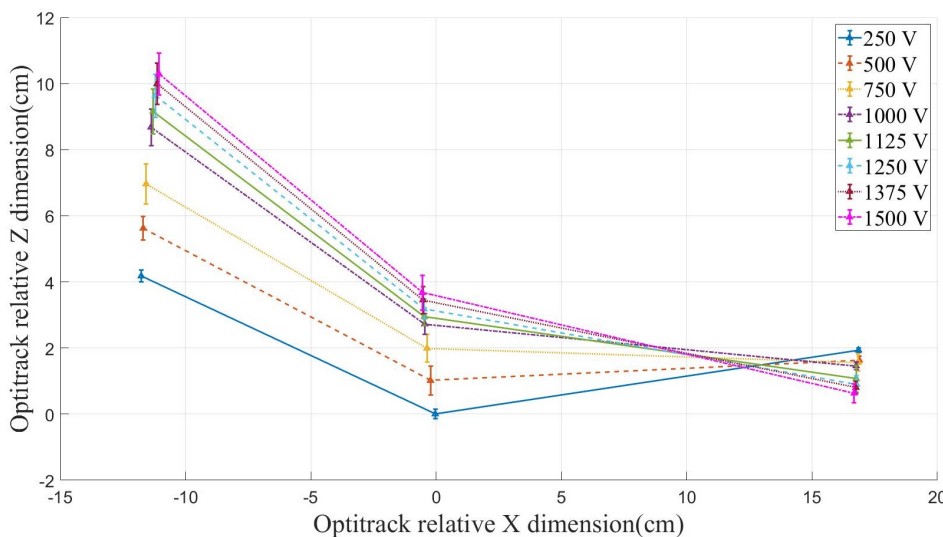

**Figure 15.** Tail twist repeatability experiments: Frontal view.

This graphic has similar accuracy to the previous one. That situation guarantees the capabilities of the model to be controlled. The negative deformation of the left actuator does not have a negative influence on the left actuator, which makes it possible to perform a deformation of 20 degrees in the front plain of the system. For that reason, the tail can perform enough actuation to control the Roll of the ornithopter. In this case, the wind applied has less impact in the tail displacement. This is explained by the different geometry that is caused by this displacement. In the previous case of a tail raise, all the tail's surface is opposed to the wind. However, in this torsion experiment, both sides of the tail move in different direction, allowing the wind to flow around the tails surface in an easier way. Tail deformation is maintained, as in the precious experiment.

In conclusion to the physical aspect of the tail, the system is a good solution. Our previous experience controlling and analyzing with servos generated in works like [47,48], generated limits in the Roll and Pitch of the tail to control the ornithopter. These limits are surpassed by the system that is presented in this paper. The system reduces the mass, and the capability to be deformed improves the capability of controlling the system. Additionally, the system is easy to control by modeling the voltage inputs of the system.

### 4.4. Outdoors Experiments

In this section, the outdoor experiments are showed and analyzed. Figure 16 shows the ornithopter setup for these experiments. In these tests, the ornithopter made for the *GRIFFIN* project is used. The ornithopter setup includes the electronics that are presented in Section 3.3 to control the bio-inspired tail in different configurations. Additionally, the distribution of the mass is centered in the center of gravity of the ornithopter.

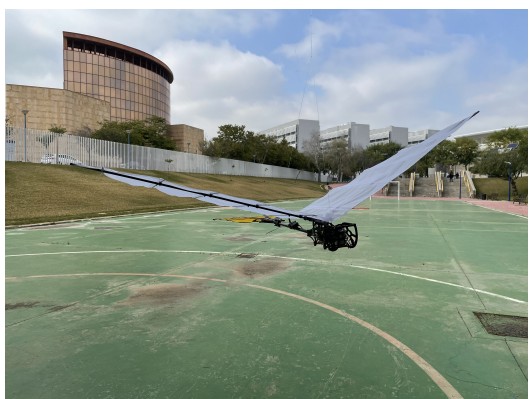

**Figure 16.** Outdoor Setup.

The experiments were performed outdoors in the localization 37°24′39.4′′ N 6°00′10.4′′ W on a sunny day. Two kinds of experiments were performed. Firstly, the capacity of the tail to perform a turn on the ornithopter trajectory is proved. Figure 17 shows a representation of this experiment. Secondly, the capacity of the tail to control the trajectory of the ornithopter is proved in Figure 18.

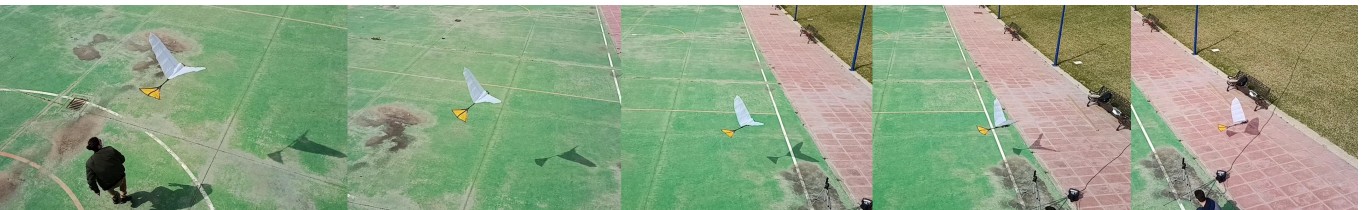

**Figure 17.** Twist Flapping-Flight Composition Test. Top View.

Figure 17 shows the ornithopter capabilities to turn using the bio-inspired morphing tail. In this experiment, the left actuator was fully actuated to perform a turn in the right direction. The ornithopter performs a change in the direction of 180 degrees in two meters longitude. This is an excellent result, proving that the tail can provide enough actuation to change the ornithopter trajectory.

The next step is to prove that the actuation can be used to control the flight trajectory. For this reason, an experiment performing a manually controlled straight line was performed. Figure 18 shows the results of these experiments.

The result of this experiment shows that it is possible to control the trajectory using the tail that was developed in this work. The reduced mass of the tail that is presented in this work improves the capabilities for controlling the system. Additionally, the tail

actuation based on nature using a non-linear aerodynamics profile improves the capabilities to stabilize the ornithopter trajectory.

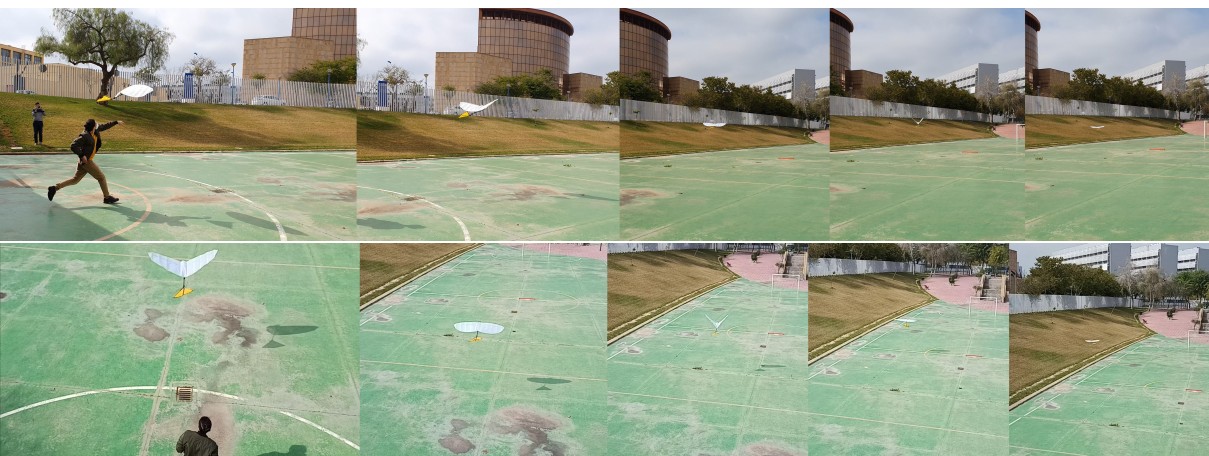

**Figure 18.** Manually Controlled Straight Flapping-Flight Composition.

## 5. Conclusions

In this paper, a new concept of a bio-inspired tail and a new method of actuation was presented.

The tail is capable of performing morphing using a bi-morph configuration of Macro Fiber Composites (MFC).

The actuation method provides a new horizon in the aerial robots field. Robots that can change their shape themselves, performing morphing, in order to generate actuation on their trajectory.

In the future, these actuation methods will be used on different surfaces of the system. One of the most interesting surfaces to apply this method to the ornithopters will be the wings. The objective of using this actuation system on the wing is to improve control capabilities. In ornithopters, the control capability is limited due to the geometry of the aerial system itself; for this reason, using systems that improve this capability without changing the general geometry of the robot will be a huge step in aerial robotics development.

It has been shown that the tail concept works having high repeatability and accuracy in its behavior. The repeatability and accuracy of the system, including wind conditions, make it possible to elaborate a control law to control the system. Additionally, the displacement of the actuation, which has been a limitation in previous works, is enough for considering it to be a real alternative to traditional actuators used in ornithopters. The results obtained in this work will be the key for developing a reliable control that will allow a real time trajectory control of the flapping-wings aerial robot using bio-inpired approaches.

Several experiments and analyses of the concept have been done, showing excellent results. The continuity of the displacement has been successfully measured using a Mocap system and showing a maximum displacement of nearly 80 mm of linear displacement and 20° of rotary.

The outdoors experiments proved that the system performs a significant improvement on the current tail system. These experiments show excellent results, making the control of the ornithopter with a MFC bio-inspired morphing tail possible.

Therefore, it is possible to state that the concept presented in this work is capable to perform morphing, trying to mimic animals' behavior, and performing the displacement of the tail that is needed for the full control of an ornithopter.

Future works will focus on two main directions. Firstly, the development of a second version of the tail improving the capabilities of the first version. The idea is to maximize the blocking force, maintaining the displacement and improving the design and manufacturing process.

Secondly, a better understanding of the properties of these materials and their capabilities have been achieved due to the theoretical and practical analysis of the MFC done in this work. Hence, the idea will be to explore other applications of these piezo-electrics in addition to the tail. One potential application, as exposed previously, is the morphing of the wings of the ornithopter, which would allow changing the aspect ratio or other wing properties, depending on demand, so that the efficiency and performance of the UAS could be increased.

The aim is to implement soft-robotic and bio-inspired solutions in the aerial systems for improving their capabilities.

**Author Contributions:** Conceptualization, V.P.-S., A.E.G.-T. and B.C.A.; methodology, V.P.-S., A.E.G.-T. and B.C.A.; software, V.P.-S., A.E.G.-T. and E.S.; validation, V.P.-S., A.E.G.-T. and E.S.; formal analysis, V.P.-S. and A.E.G.-T.; investigation, V.P.-S. and A.E.G.-T.; resources, V.P.-S. and A.E.G.-T.; data curation, V.P.-S., A.E.G.-T. and E.S.; writing—original draft preparation, V.P.-S., A.E.G.-T. and E.S.; writing—review and editing, V.P.-S., A.E.G.-T., E.S., B.C.A. and A.O.; visualization, V.P.-S. and A.E.G.-T.; supervision, B.C.A. and A.O.; project administration, A.O.; funding acquisition, A.O. All authors have read and agreed to the published version of the manuscript.

**Funding:** This work has been funded by the European Research Council Advanced Grant GRIFFIN (General compliant aerial Robotic manipulation system Integrating Fixed and Flapping wings to INcrease range and safety), Action 788247.

**Institutional Review Board Statement:** Not applicable.

**Informed Consent Statement:** Not applicable.

**Acknowledgments:** We thank *Robotics, Vision and Control Group* for supporting us during this work. We also thank our colleagues for their help in the experiments.

**Conflicts of Interest:** The authors declare no conflict of interest.

## Abbreviations

The following abbreviations are used in this manuscript:

| | |
|---|---|
| MDPI | Multidisciplinary Digital Publishing Institute |
| DOAJ | Directory of open access journals |
| TLA | Three letter acronym |
| LD | Linear dichroism |
| MFC | Macro Fiber Composites |
| UAS | Unmanned Aerial System |
| TCP | Twisted and Coiled Polimers |
| UAV | Unmanned Aerial Vehicle |
| MAV | Micro Aerial vehicles |
| CLT | Classical Lamination Theory |
| Mocap | Motion Capture System |

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
