# Peer review of "Bio-Inspired Morphing Tail for Flapping-Wings Aerial Robots Using Macro Fiber Composites"

_applsci, doi:10.3390/app11072930_

Round 1

Reviewer 1 Report

Comments:
- in figure 7, pictures a, b should be signed.
- line 409: three times you used "of" in one sentence . This sentence should be improved. 
- Figures 8,9: some colors used overlap - it should be improved. Maybe you can use additional mark. Ordinate and abscissa - change signatures. They are the same, maybe you can use coordinate system. In "d" picture the number 30 and 0 overlapped. Why in picture 8d you have position from 0 to 8cm, but in figure 9d you have position from 0 to 18cm? Is it the same model?
- Figure 13 - mistake with number of pictures, lack of "e", but you have a "g". Where is the coordinate system? Where is X, Y and Z? You should give this coordinate system in picture with model. You present results  with 1V and 6V, but your description is about 250V and 1500V. Is it correct? 
- Figure 14,15 - font too small on the axes. Why the graph is not starting from 0? Where is 0? From what points are these results? It should be described. 
- line 576: 100 experiments ? where are these results? It should be corrected. 
- Figures 17,18 - how do we know there was no wind and the model changed direction because of this? :) these drawings are not objective. 
General comment:
It is very interesting work, but only focused on experiment. There is no confirmation that what you are showing is true since it cannot be calculated numerically or theoretically. If it possible, any validation of experience would be appreciated. 
How will such a model behave in the rain? Will it be able to work with the same efficiency in rain? Have you checked it? In summary, you write that such a structure can be controlled. Are you planning any research in this direction? 
I know one team from Lublin University of Technology (Poland) deals with the control of structures with MFC elements. The paper (https://www.sciencedirect.com/science/article/pii/S0263822319337766) presents experimental and numerical (FEM) tests on composite structures. Maybe they will be helpful in conducting numerical tests of your model.

Reviewer 2 Report

In this manuscript, the authors designed and developed a bioinspired morphing tail for ornithopters. The bioinspired tail used piezoelectric-based macro fiber composites (MPCs) as the actuating components, which demonstrated multiple advantages over traditional servomotor-based systems, including lower power consumption, smaller weight, speed, and environmental resistance. The authors also provided field experiments and showed that the MPC-tailed ornithopter could change its trajectory during flights. Overall, this is a nicely written paper with a comprehensive overview of previous works in the field. Below are a few thoughts: (1) It would be good to elaborate on the bioinspiration in the introduction, for instance, what tail structures birds have evolved and what approaches they take for maneuver and energy saving during flights. I think this would tie closely to the bioinspired claim of the paper. (2) It would be good to include epoxy components in the schematic of Figure 2. (3) The authors mentioned that the power consumption of their piezoelectric MFC actuator is almost zero. Would there be any possible energy dissipation in actual conditions? (4) Could the authors make a more quantitative comparison on the maximum external force that either an MPC-integrated tail or a traditional servomotor-powered V tail can withstand? If there is a large external force, would the MPC tail still be able to actuate properly with the same performance? (5) I noticed that the actual operating voltage to actuate the MPC tail is ~ 1500 V. Would the authors suggest any ways to further reduce the voltage? (6) The authors showed two very nice demonstrations of outdoor flight controls. How would the MPC-integrated ornithopter function under windy conditions?
